# Predicting antigen specificity of single T cells based on TCR CDR3 regions

David S Fischer[1,2] (ID), Yihan Wu[1] (ID), Benjamin Schubert[1,3] (ID) & Fabian J Theis[1,2,3,*] (ID)

## Abstract

It has recently become possible to simultaneously assay T-cell specificity with respect to large sets of antigens and the T-cell receptor sequence in high-throughput single-cell experiments. Leveraging this new type of data, we propose and benchmark a collection of deep learning architectures to model T-cell specificity in single cells. In agreement with previous results, we found that models that treat antigens as categorical outcome variables outperform those that model the TCR and antigen sequence jointly. Moreover, we show that variability in single-cell immune repertoire screens can be mitigated by modeling cell-specific covariates. Lastly, we demonstrate that the number of bound pMHC complexes can be predicted in a continuous fashion providing a gateway to disentangle cell-to-dextramer binding strength and receptor-to-pMHC affinity. We provide these models in the Python package TcellMatch to allow imputation of antigen specificities in single-cell RNA-seq studies on T cells without the need for MHC staining.

**Keywords** antigen specificity; multimodal; single cell; supervised learning; T-cell receptors

**Subject Categories** Computational Biology; Immunology

**Mol Syst Biol. (2020) 16: e9416**

## Introduction

Antigen recognition is one of the key factors of T cell-mediated immunity. T cells interact via a dimeric surface protein, the T-cell receptor (TCR), with an antigen presented on a major histocompatibility complex (MHC) located on the surface of antigen-presenting cells. This presenting cell can be experimentally modeled via an MHC multimer with an immobilized antigen (pMHC). The T cells of an individual organism cover a wide range of antigen specificities. This variability in specificity stems mostly from plasticity of three complementarity-determining region (CDR) loops (CDR1-3) of both TCR α- and β-chains. The hypervariable loops CDR3α and CDR3β are most commonly aligned with the presented epitope (Singh *et al*, 2017) and are hypothesized to

be the main driver of T-cell specificity (Glanville *et al*, 2017). However, specificity-determining influences of the other CDR loops (Cole *et al*, 2009; Madura *et al*, 2013; Stadinski *et al*, 2014) and distal regions (Harris *et al*, 2016a,b) have also been demonstrated.

The ability to accurately predict T-cell activation upon antigen recognition based on antigen and TCR sequences would have transformative effects on many research fields from infectious disease, autoimmunity, and vaccine design to cancer immunology, but has been thwarted by a lack of training data and adequate models. In the absence of sufficiently large experimental data, most studies focused on molecular analysis of individual co-crystalized TCR–pMHC complexes and molecular dynamics simulations with limited success (Flower *et al*, 2010). Only recently, through concerted data collection efforts (Borrman *et al*, 2017; Shugay *et al*, 2018; Vita *et al*, 2019) and newly emerging high-throughput technologies that allow the sequencing of the TCR while probing the T-cell specificity (Klinger *et al*, 2015; Bentzen *et al*, 2016), have large enough data sets become available to begin modeling the TCR–pMHC interaction through machine-learning methods (Zvyagin *et al*, 2020). Current methods to predict the likelihood of binding of TCRs to specific antigens use linear position-specific scoring matrices (Glanville *et al*, 2017), Gaussian processes (preprint: Jokinen *et al*, 2019), or random forests (Gielis *et al*, 2018). A second set of methods attempts to directly model the TCR–pMHC interaction with neural networks in order to generalize across unseen TCR–antigen pairs (preprint: Jurtz *et al*, 2018). We expand on these efforts but also consider the current limitation in the number of available antigens in training data sets. Secondly, we consider the inclusion of complex sets of cell-specific covariates into the prediction problem. The inclusion of cell-specific covariates has previously been shown to work in the example of transcriptome-derived clusters as covariates (preprint: Jokinen *et al*, 2019). Here, we leverage the data modalities in the new droplet-based single-cell experiments.

In this study, we exploit a newly developed single-cell technology that enables the simultaneous sequencing of the paired TCR α- and β-chains and determining the T-cell specificity via bound peptide-loaded MHC (pMHC) complexes. This technology allows the routine collection of binding TCR and antigen complexes of the size of entire curated databases in a single study (Bagaev *et al*, 2019; 10x Genomics, 2019) and accordingly harnesses great potential to transform the field of T-cell receptor specificity prediction. We propose and trained multiple deep learning architectures that model the TCR–

1 Institute of Computational Biology, Helmholtz Zentrum München, Neuherberg, Germany
2 TUM School of Life Sciences Weihenstephan, Technical University of Munich, Freising, Germany
3 Department of Mathematics, Technical University of Munich, Garching bei München, Germany
*Corresponding author. Tel: +49 89 3187 43260; E-mail: fabian.theis@helmholtz-muenchen.de

pMHC interaction. The models account for the variability found in single-cell data through cell-specific covariates. We show that models that include both α- and β-chain have a predictive advantage over models that only include the β-chain, while models fit on only a single chain still perform well. We further find that T-cell specificity imputation in a single-cell sample from a known donor is possible, enabling assessment of the presence of disease-specific T cells, while generalization across unknown TCR–pMHC pairs is still not possible. Lastly, we anticipate a large number of single-cell studies involving T cells to exploit TCR specificity as an additional phenotypic readout. To facilitate the usage of our predictive algorithms, we built the Python package *TcellMatch*, which hosts a pre-trained model zoo for analysts to impute pMHC-derived antigen specificities and allows the transfer and re-training of models on new data sets.

## Results

### A joint deep learning model for alpha- and beta-chains, antigens, and covariates for single-cell TCR profiling experiments

We set out to predict the antigen specificity of single T cells based on TCR α- and β-chain sequences and other cellular covariates, such as donor identity and cell surface protein counts. We used a publicly available single-cell data set (10x Genomics, 2019) based on a technology in which cells are captured in droplets in a microfluidics system so that antigen specificity, the CDR3 TCR sequences, surface protein abundance, and mRNA abundance can be assayed for each captured cell (Fig 1A, Methods and Protocols). Antigen specificity was quantified via the count of unique molecular identifiers associated with antigen-specific dextramer (pMHC complex) barcode sequences (10x Genomics, 2019). Additionally, we used databases (IEDB; Shugay *et al*, 2018; Vita *et al*, 2019) and VDJdb (Shugay *et al*, 2018) that harbor additional pairs of binding TCR and antigen

sequences from traditional low-throughput screenings and crystal structures to validate our results. The prediction of antigen specificity was previously attempted on smaller data sets, but the new single-cell technology enables the collection of data sets that are orders of magnitude larger than what was previously available from curation efforts that integrated studies from the entire field of TCR specificity (Shugay *et al*, 2018; Vita *et al*, 2019). These large single-cell data sets may, however, be susceptible to greater noise than results derived from studies that are either conducted in bulk or validated separately. We chose deep learning models for the prediction task as these are well suited to cope with large noisy data sets. We included interpretable linear models and a previously proposed non-linear reference model (*NetTCR*; preprint: Jurtz et al, 2018) as baseline methods. The convolutional and linear models used here are in structure similar to models that relate antigen specificity to clusters of TCR sequences but are continuously differentiable and therefore easier to extend to new specificity groups.

The prediction of antigen specificity from TCR sequences and numeric cellular covariates is a mixed input data-type problem. The deep characterization of the single cells via modalities such as mRNA or surface protein abundance in the context of specificity assessment makes such mixed input data-type models much more relevant to single-cell data than they were previously to less well-characterized pairs of binding TCRs and antigens that were curated from literature. We approached this problem by combining a network tailored to numerical data with a network tailored to sequence-structured data to yield a single prediction (Fig 1B). Machine learning on sequence data is a field of ongoing research and different layer types have been shown to be effective for different tasks. Accordingly, we implemented all major sequence data-specific layer types to be able to perform a comprehensive comparison of deep learning architectures for the task of predicting TCR specificity. This comprehensive comparison is to the best of our knowledge the first of its kind. Specifically, we implemented recurrent layers (bidirectional GRUs; Schuster &

**Figure 1. Deep learning models predict binding of T-cell receptors (TCR) to peptide MHC complexes (pMHC) from defined antigen panels.**

Distributions shown as boxplots are across threefold cross-validation. *AUC ROC test*: Area under the receiver operating characteristic curve on the test set for the binary binding event prediction task. The top panel in (C), (F), (G) is a zoom into an informative region of the *y*-axis. *counts*: total mRNA counts, *nc*: negative-control pMHC counts, *surface*: surface protein counts.

A Concept of multimodal single-cell immune profiling experiment with RNA-seq, surface protein quantification, bound pMHC quantification, and TCR reconstruction.

B Categorical *TcellMatch* model: A feed-forward neural network to predict a vector of antigen specificities of a T cell based on the CDR3 sequences of the TCR α- and β-chains. *Gray boxes*: layers of the neural network.

C Covariates improve sequence-based binding accuracy prediction. Shown are bidirectional GRU models fit on both α- and β-chains (CONCAT). *none*: no cell-specific covariates, *donor*: one-hot encoded donor identity, *donor + counts*: one-hot encoded donor identity and total mRNA counts per cell, *counts*, *nc*: negative-control pMHC count vector, *nc + donor + counts*: negative-control pMHC count vector, one-hot encoded donor identity and total mRNA counts per cell, *counts*, *nc + donor + counts + surface*: negative-control pMHC count vector, one-hot encoded donor identity, total mRNA counts per cell and surface protein count vector (*n* = 4 cross-validations for models *none* and *nc*, "leave-one donor out", and *n* = 3 cross-validations for all other models).

D Overlap of correctly and incorrectly classified test set observations from best-performing model to models with reduced covariate sets. Models without donor covariates were not included. *full*: nc + donor + counts + surface model from (C), *red*: model shown on x-axis tick (*n* = 3 cross-validations for all models).

E Antigen-wise prediction performance by covariates setting. In contrast to panel (C), the prediction performance is not aggregated across the entire test set but evaluated separately the observations belonging to each antigen. Shown are bidirectional GRU models fit on both α- and β-chains (CONCAT) (*n* = 4 cross-validations for models without donor covariate, "leave-one donor out", and *n* = 3 cross-validations for all other models).

F Antigen-binding prediction is improved by the inclusion of TCR CDR3 sequences. *BIGRU*: bidirectional GRU model, *NOSEQ*: model without TCR sequence embedding. Models without donor covariates were not included (*n* = 4 cross-validations for models *none* and *nc*, "leave-one donor out", and *n* = 3 cross-validations for all other models).

G Antigen-binding prediction based on TCR CDR3 sequences is improved by modeling α- and β-chains. *BIGRU*: bidirectional GRU model, *SA*: self-attention model, *CONV*: convolution model, *LINEAR*: linear model, *CONCAT*: models fit on the CDR3 sequences of both TCR α- and β-chains, *TRA*, *TRB*: models fit on the CDR3 sequence of either the TCR α- or the β-chain (*n* = 3 cross-validations for all other models).

Data information: All boxplots: the center of each boxplot is the sample median; the whiskers extend from the upper (lower) hinge to the largest (smallest) data point no further than 1.5 times the interquartile range from the upper (lower) hinge. In (C, F, G), the underlying data points are shown as swarm plots color-coded in the same way as the boxplot.

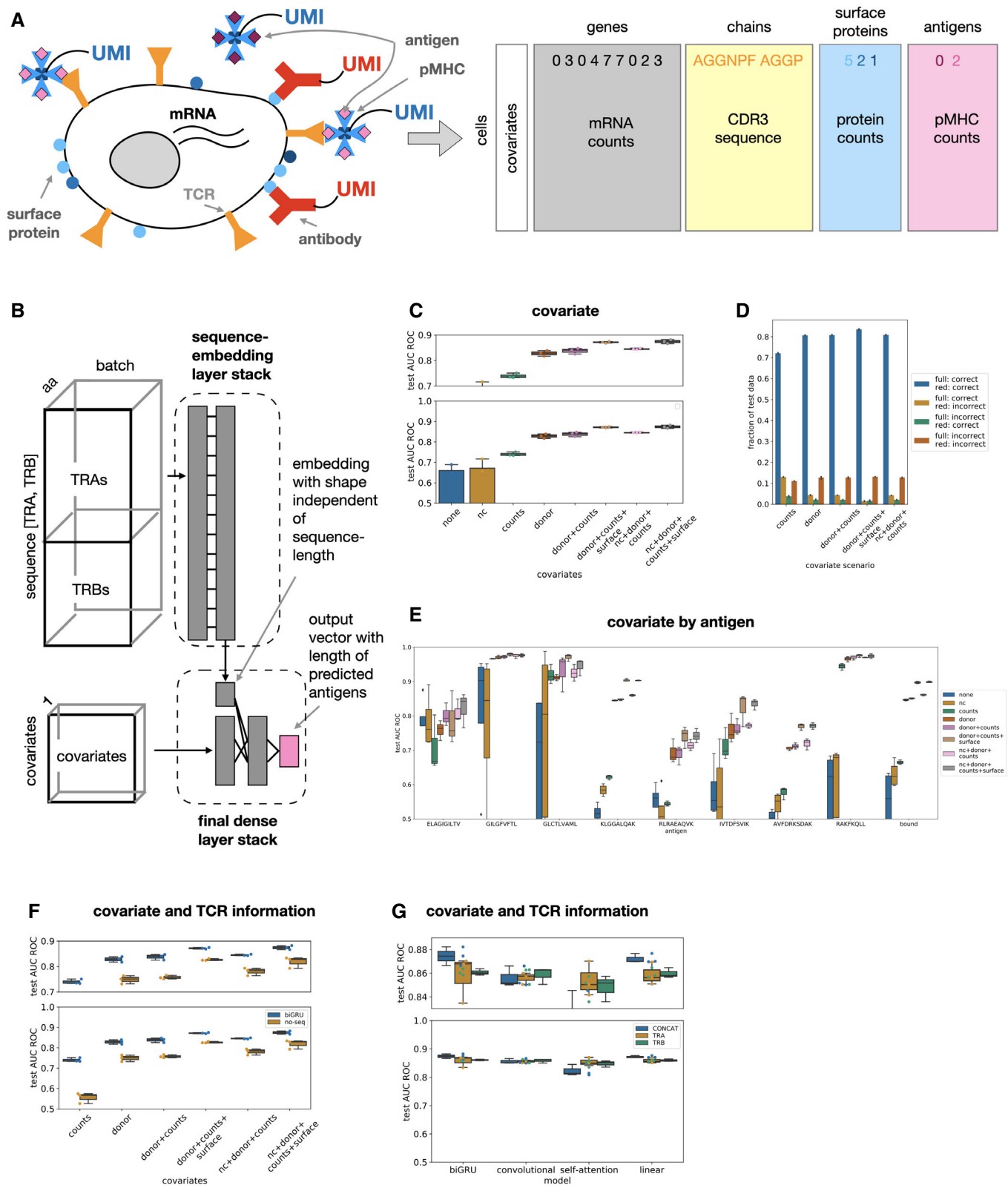

**Figure 1.**

Paliwal, 1997; Cho *et al*, 2014) and bidirectional LSTMs (Hochreiter & Schmidhuber, 1997; Schuster & Paliwal, 1997), convolutional layers (Szegedy *et al*, 2015), self-attention layers (Vaswani *et al*, 2017), and densely connected networks, which include linear models that relate to previous work (Glanville *et al*, 2017). All of these sequence data embedding layer types require an initial representation of the elements

of the sequence: an initial encoding of the amino acids. We compared categorical, substitution frequency derived (BLOSUM), and learned embeddings and found that the initial amino acid embedding does not have a strong effect on the results (Appendix Fig S1). The novel learned embedding that we propose here is more parameter efficient as it can expose a lower-dimensional amino acid space to the sequence-embedding layers than the standard embedding layers do (Methods and Protocols). In the following, we only show model fits based on these learned 1×1 convolutional embeddings based on BLOSUM50 (Methods and Protocols).

We considered the binding event prediction task within a panel of antigens as a single- or multi-task prediction problem with antigen species as categorical output variables ("categorical antigen model", Figs 1 and 2, Methods and Protocols). Secondly, we considered binding event prediction on arbitrary antigens as a distinct scenario that requires the model to embed the input antigen sequence ("antigen-embedding model", Fig 3, Methods and Protocols). The categorical antigen model predicts a probability distribution across possible binding events, including a negative (no binding) event. The antigen-embedding model is based on the concept of positive and negative sets. In the single-cell data, a negative set naturally arises from cells that did not bind to any or a given pMHC species. The positive set is naturally defined as the observed binding pairs. We generated the negative set for TCR–antigen-binding pairs from IEDB or VDJdb (preprint: Jurtz *et al*, 2018) shuffling TCR and antigen assignments *in silico*.

## Assembling meaningful training and test sets across databases

We subset the data sets to allow a meaningful model comparison and predictivity evaluation: The single-cell data set contained more than 150,000 cells from four donors with successfully reconstructed TCR sequences and with measured binding specificity to 44 distinct pMHC complexes. The authors of this data set defined binding events by comparing the target pMHC counts to the counts of negative-control pMHCs. pMHCs were defined as negative-control pMHCs if they were not expected to specifically bind any TCR in the screen (10x Genomics, 2019). We assembled antigen specificity labels based on the same binding classification scheme. We removed putative cellular doublets from the data set (Methods and Protocols, Appendix Fig S2): A doublet of two cells of distinct specificities in a microfluidics setup may result in the TCR sequence of the first cell and the pMHC binding read-outs from the second cell being misreported as a third, non-existent, specificity pair. To avoid such non-existent specificity pairs, we chose a conservative doublet exclusion threshold (Methods and Protocols). We only considered the eight antigens in the pMHC CD8$^+$ T-cell data set that had at least 100 unique, non-doublet clonotype observations to remove effects from strong class imbalance (Appendix Fig S3A and B). The total data set size was 91,495 unique, non-doublet observations (cells) across the four donors.

We only assembled pairs of binding TCR CDR3 β-chain and antigen sequences from IEDB and VDJdb as these databases contain far fewer α-chain than β-chain sequences and do not contain an equivalent of the cellular covariates found in the single-cell data. We only considered observations from the most commonly assayed HLA type HLA-A*02:01. We assembled a data set of 12,414 observations from 10,726 clonotypes and 71 antigens from IEBD and 3,964 observations from 2,812 clonotypes and 40 antigens from VDJdb, which contained

at most 10 TCR sequences per clonotype. The number of TCR clonotypes per antigen was very heterogeneous, with the most frequently encountered antigen covering 4,812 clonotypes in IEDB, and 1,461 in VDJdb. We provided a detailed descriptive analysis of all data sets in Dataset EV3. TCR and specificity variation of the single-cell data are also described in detail elsewhere (10x Genomics, 2019).

To avoid an over-optimistic estimation of model performance, we clustered the T cells into clonotypes and separated the single-cell data into train, test, and validation sets with regard to their assigned clonotypes so that each clonotype only existed in one of the splits (Methods and Protocols). We down-sampled clonotypes to a maximum of 10 observations.

## Cell-specific covariates improve binding event prediction

Single-cell T-cell specificity screens feature multiple effects that confound the binding event and its observation. Here, we compared the performance of categorical antigen models with various sets of covariates to quantify the relevance of covariates for predictive models.

Firstly, one would expect the donor identity to affect the TCR sequence if donors vary in their HLA genotype. We compared models with and without a one-hot encoded donor identity covariate to establish the impact of these donor-to-donor differences. We found that the performance of models without donor information varies strongly and is much worse than the performance of models with donor covariates. The mean area under the receiver operating characteristic curve (AUC ROC, Methods and Protocols) was 0.33 for bidirectional GRU models (the best-performing sequence-based models) without covariates and 0.81 for those with donor covariates (Fig 1C).

The identification of binding events based on single-cell RNA-seq libraries is liable to false negatives due to a low capture rate of RNAs. In the single-cell screen, negative-control pMHCs were included to provide a background distribution of non-specific binding events and were part of the definition of discrete binding events (Methods and Protocols). The discrete labels are therefore already corrected for false-positive binding events. We investigated whether normalization factors and negative-control pMHC counts are useful predictors of a false-negative binding event that cannot be rescued by background signal correction: A donor covariate-only model ("donor") was not outperformed either by a model that also included a scaled total mRNA count covariate ("donor + counts") or by one that additionally also contained negative-control count covariates ("nc + donor + counts") (Materials and Methods, Fig 1C). We conclude that such false-negative observations are either rare or cannot be captured by the correction proposed here. We also identified a predictive advantage of models that account for the cell state encoded by surface protein counts: bidirectional GRUs that accounted for donor, negative-control pMHC counts, and total counts improved from 0.83 AUC ROC to 0.86 if cell surface protein counts were added as a covariate (Fig 1C, Welch's *t*-test, $P < 0.01$). The surface protein counts can be used to embed cells based on their membrane surface structure in a latent space which can be used by the model to account for the abundance of TCRs and other binding-relevant proteins on the cell surface. The overall top-performing model accounted for donor, total counts, negative-control counts, and surface protein counts with an AUC ROC of 0.87 (Fig 1C).

We validated that growing the set of covariates modeled lead to models that had additional (rather than different) correct

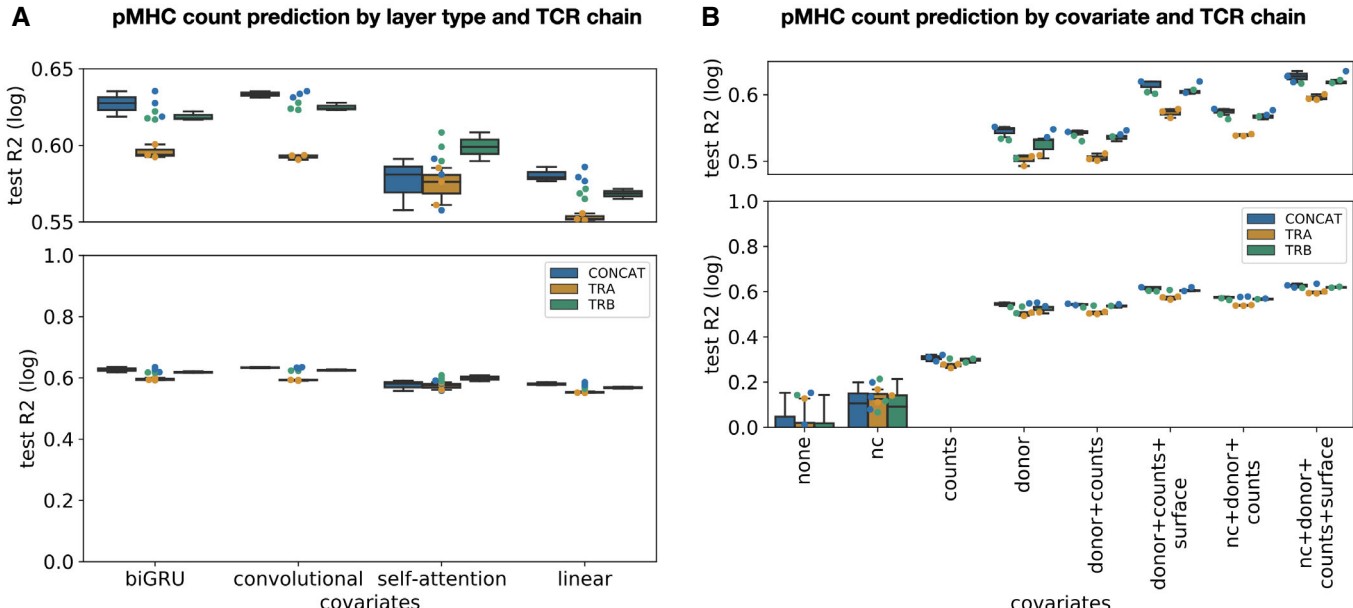

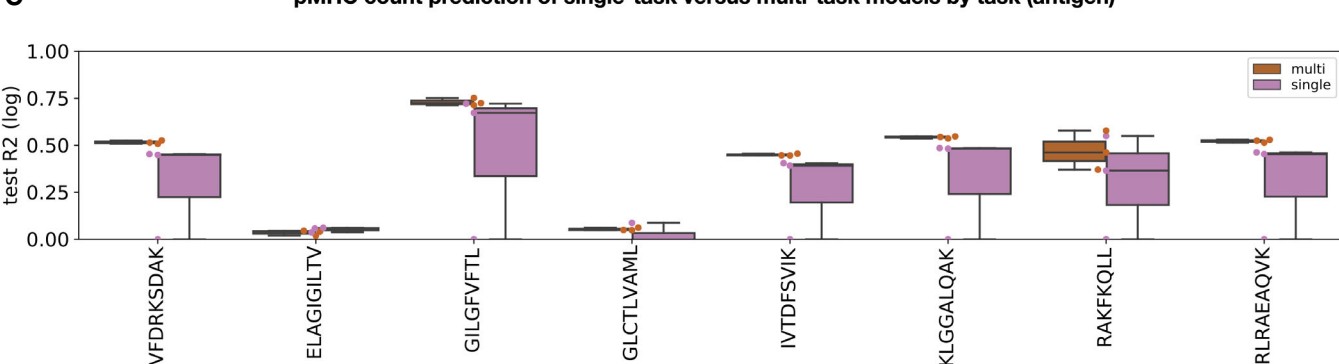

**Figure 2. The binding strength of T cells to pMHC complexes can be modeled based on single-cell data.**

A  Sequence-encoding layer types outperform linear models on pMHC count prediction if donor and size factors are given as covariates. *BIGRU*: bidirectional GRU model, *SA*: self-attention model, *CONV*: convolution model, *LINEAR*: linear model, *CONCAT*: models fit on the CDR3 sequences of both the TCR α- and β-chains, *TRA*, *TRB*: models fit on the CDR3 sequence of either the TCR α- or the β-chain (n = 3 cross-validations for all other models).

B  Performance of bidirectional GRU models that predict pMHC counts directly is best if covariates and both TCR chains are modeled. *test R2 (log)*: test R2 on log-transformed test data. *none*: no cell-specific covariates, *donor*: one-hot encoded donor identity, *donor + counts*: one-hot encoded donor identity and total mRNA counts per cell, *counts*, *nc*: negative-control pMHC count vector, *nc + donor + counts*: negative-control pMHC count vector, one-hot encoded donor identity and total mRNA counts per cell, *counts*, *nc + donor + counts + surface*: negative-control pMHC count vector, one-hot encoded donor identity, total mRNA counts per cell and surface protein count vector (n = 4 cross-validations for models without donor covariate, "leave-one donor out", and n = 3 cross-validations for all other models).

C  Multi-task models outperform separate single-task model on pMHC count prediction by antigen. *multi*: multi-task model, *single*: single-task model (n = 3 cross-validations for all other models).

Data information: All boxplots: the center of each boxplot is the sample median; the whiskers extend from the upper (lower) hinge to the largest (smallest) data point no further than 1.5 times the interquartile range from the upper (lower) hinge. The underlying data points are shown as swarm plots color-coded in the same way as the boxplot.

predictions. The best-performing model with the highest number of covariates predicted almost all observations correctly that were also predicted correctly by models with fewer covariates (Fig 1D). The test sets are not balanced across the different classes: We found similar trends across covariate settings on each individual class as we found globally (Fig 1E). We validated that sequence information

is indeed a relevant predictor in each of these covariate scenarios, indicating that the combination of sequence and non-sequence covariates is desirable (Fig 1F).

Lastly, we investigated whether models that were fit with cell-specific covariates generalize to observations that do not contain these covariates. For this purpose, we applied models presented in

this section to TCR sequences from matched and unmatched antigens from IEDB (Vita *et al*, 2019) and VDJdb (Shugay *et al*, 2018), setting the covariate input vector to zero. The best-performing linear predictors had true-positive rates above 0.55 while maintaining false-positive rates below 0.1 (Appendix Fig S4), suggesting that these models can generalize to settings in which not all covariates are observed.

## Co-modeling alpha- and beta-chains improves binding event prediction

We compared the predictivity of models fit using one TCR CDR3 chain ("TRA only", or "TRB only") with models fit on both TRB and TRA chains ("TRA + TRB", Materials and Methods) to evaluate the additional information inherent in the use of both chains. We found

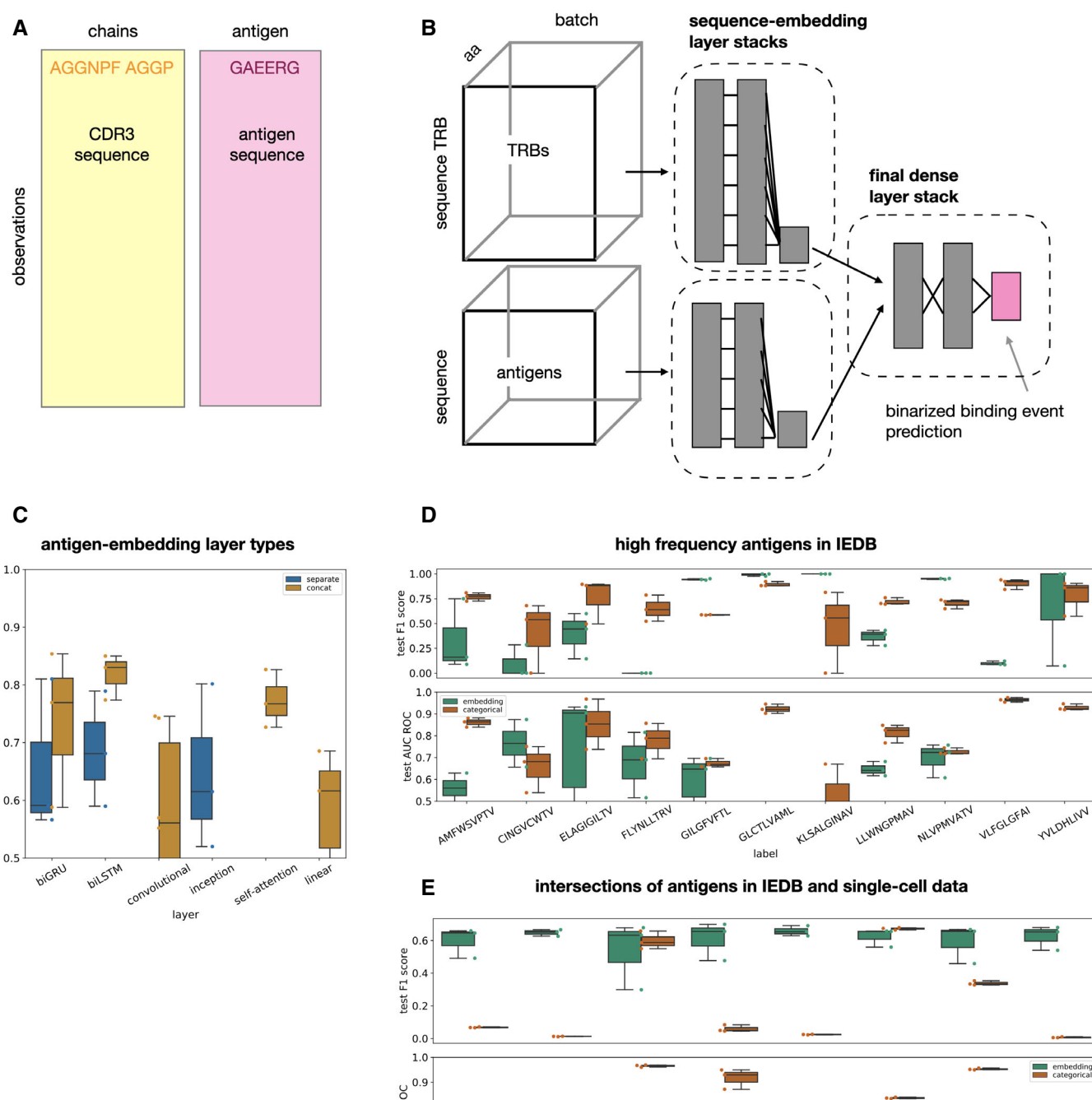

**Figure 3.**

**Figure 3. Models tailored to generalize to unseen antigens are outperformed by categorical antigen models on seen antigens.**

Distributions shown as boxplots are across threefold cross-validation.

A    The databases IEDB and VDJdb contain pairs of TCRs and antigens that were found to be specific to each other and are curated from many different studies. A supervised model that predicts binding events can be trained on such data but also requires the assembly of a set of negative observations (Methods and Protocols).

B    Antigen-embedding *TcellMatch* model: A feed-forward neural network to predict a binding event based on TCR CDR3 sequences and antigen peptide sequence. *Gray boxes*: layers of the neural network.

C    Different sequence-encoding layer types perform similarly well on binding prediction based on TRB-CDR3 and antigen sequence. *CONCAT*: models in which TRB CDR3 sequence and antigen sequence are concatenated, *SEPARATE*: models in which TRB CDR3 sequence and antigen sequence are embedded by separate sequence-encoding layer stacks. *BILSTM*: bidirectional LSTM model, *BIGRU*: bidirectional GRU model, SA: self-attention model, *CONV*: convolution model, *INCEPTION*: inception-type model, *NETTCR*: NetTCR model (preprint: Jurtz *et al*, 2018), *LINEAR*: linear model (*n* = 3 cross-validations for all other models).

D, E    Antigen-wise categorical models outperform models that are built to generalize across antigens on high-frequency antigens in IEDB (D) and on overlapping antigens between IEBD and single-cell data (E). In both cases, the models were trained on IEDB and tested on held-out observations from IEBD (D) or on the single-cell data (E). *embedding*: models that are embedding the antigen sequence and can be run on any antigen (Fig 3b), *categorical*: antigen-wise categorical models that do not have the antigen sequence as a feature (Fig 1B) (*n* = 3 cross-validations for all other models).

Data information: All boxplots: the center of each boxplot is the sample median; the whiskers extend from the upper (lower) hinge to the largest (smallest) data point no further than 1.5 times the interquartile range from the upper (lower) hinge. The underlying data points are shown as swarm plots color-coded in the same way as the boxplot.

that TRA + TRB models were slightly better than TRA-only and TRB-only models across most layer types if basic single-cell covariates were included in the prediction. The top-performing TRA + TRB was 0.01 AUC ROC better than the corresponding single-chain model (Fig 1G). This suggests that the evolutionary constraint on the α-chain is so strong that there is a strong correlation between the two chains, which is in line with recent results that are based on prediction performance on smaller single-cell data sets (preprint: Jokinen *et al*, 2019) and results based on TCR similarity (Lanzarotti *et al*, 2019). We found that recurrent and convolutional neural networks performed similarly to linear models, typically with a difference of up to 0.01 AUC ROC (Fig 1G). This suggests that antigen specificity of a α- and β-chain pair can be well represented as a sequence motif problem in which the sequence motif has a fixed position on the CDR3 sequence.

**Binding strength can be approximated based on pMHC counts**

In single-cell studies, antigen-binding events are measured as the number of bound pMHCs of the target antigen compared with bound negative-control pMHCs (Fig 1A). We hypothesized that one can predict not only binarized binding events but also binding strength based on the pMHC counts. The pMHC complexes used here are multimers ("dextramers") and there are typically many TCR complexes on the cell surface. Therefore, the number of bound pMHCs on a T cell is determined by a combination of the affinity of an individual TCR to a pMHC monomer and the number of possible interactions between the multimeric pMHC complex and TCR monomers on the cell surface, the compound binding strength.

We fit models that were similar in structure to the models dedicated to binarized binding event prediction on covariates and TCR CDR3 sequences (Fig 1B) to predict pMHC counts per cell (Fig 2A). We investigated whether total count and negative-control pMHC covariates explain additional variance in the data. In contrast to the discrete binding event prediction models, the labels (the pMHC counts) are no longer corrected for the negative-control background signal anymore in this scenario so that one would expect total counts and negative-control pMHC counts to influence the target pMHC counts. Indeed, the donor covariate-only model ("donor") was outperformed by a model that also included a scaled total

mRNA count covariate ("donor + counts", $R^2$ of log count difference 0.07) and one that additionally also contained negative-control count covariates ("nc + donor + counts", $R^2$ of log count difference 0.05; Materials and Methods, Fig 2B). We conclude that T cell-specific covariates can be used to fit variation in the pMHC count signal. The best-performing model included donor, total count, negative-control pMHC counts, and surface protein covariates with an $R^2$ of log counts of 0.63 (Fig 2B). The relevance of the surface state covariate beyond the background correction may be an indication of a separation of affinity (pMHC to TCR interaction) and the strength of the pMHC complex to T-cell interaction. This overall interaction strength and may depend on additional surface proteins that influence the binding event. Components of variation in both effects can likely be modeled based on the surface protein composition of the cell.

Weak binding events are not captured in the discretized binding data but may be represented in the pMHC counts. Such weak events may contain information about antigen–antigen similarities and therefore about output space correlations, which can be exploited by multi-task supervised learning. Indeed, we found that multi-task models that jointly model the prediction across antigens through shared hidden layers of the neural network architectures outperformed single-task models on six out of eight antigens modeled (Fig 2C). An alternative interpretation of the improved performance of multi-task models is their ability to learn better de-noised low-dimensional representations of TCR sequences, through the integration of more diverse training data.

**Models with sequence-space embedding of antigens are outperformed by categorical models**

The categorical approach to modeling antigens suffers from the disadvantage that predictions of unseen antigen sequences are difficult or impossible. Models that are based on a learned embedding of the antigen amino acid sequence can overcome this limitation in principle and have been used (preprint: Jurtz *et al*, 2018) to predict binding events in databases such as IEBD (Vita *et al*, 2019) or VDJdb (Shugay *et al*, 2018) (Fig 3A and B). However, it is unclear whether the antigen diversity in the currently available data is sufficient to learn such a generalization across antigens as it does not yet

adequately cover the antigen space. To resolve this issue, we first built and compared a zoo of models that can embed antigen amino acid sequences to use the best-performing instances as an upper limit on predictive performance. Entries in the IEDB and VDJdb mostly contain TCR β-chain sequences only. Accordingly, we built models that only use the TCR β-chain sequence to be able to conduct a meaningful extrapolation between the single-cell data, IEDB, and VDJdb. Previously, a specific single-layer motif-based

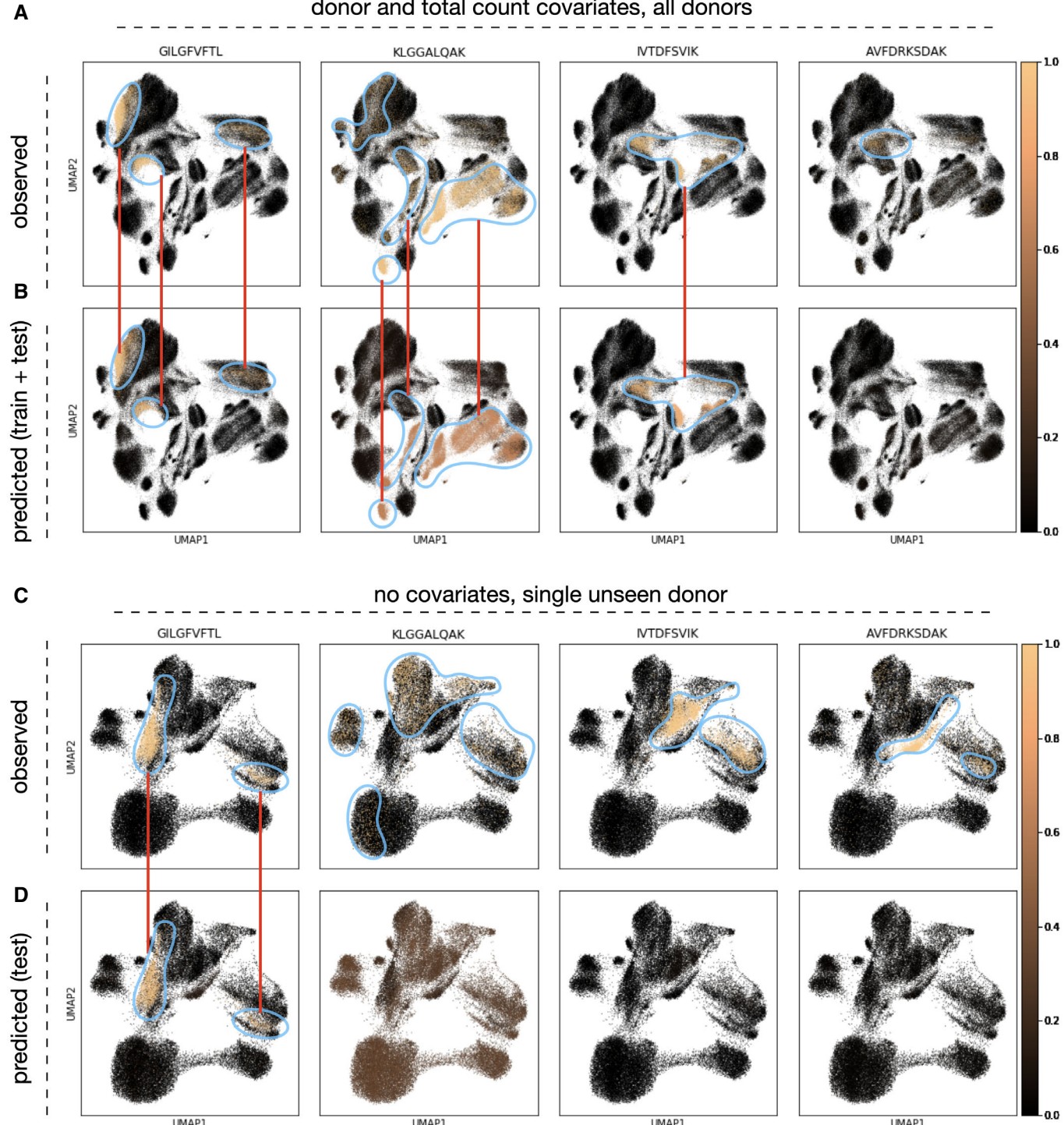

**Figure 4. Imputed antigen specificity labels enrich single-cell RNA-seq workflows on T cells by an additional phenotype.**

A–D   UMAP with observed (A, C) and predicted (B, D) labels. (A, B) The cells in the UMAP are the cells from all donors (training and validation data, $n$ = 189,512); the model was fit with donor and size factor covariates. (C, D) The cells in the UMAP are the cells from a validation donor ($n$ = 46,526); the model was fit without covariates.

architecture was proposed to model antigen sequences (preprint: Jurtz *et al*, 2018). We generalized this architecture and found that all common sequence-embedding layer types can perform this prediction and that bidirectional LSTM-based networks perform best in terms of model uncertainty with AUC ROC of 0.82 (Fig 3C). Having built optimal antigen-embedding models, we assessed whether we find evidence for the ability of these models to generalize in the antigen space on both the prediction task on antigens that are contained in the training set and the task held-out test antigens.

Firstly, we investigated whether antigen-embedding models have predictive advantages over similar categorical models on antigens that are in the training set. A lack of such predictive advantages would be indicative of an inability to learn generalizable embeddings of antigen sequences. We found that antigen-wise categorical models have better predictive performance on the antigens they were trained on than sequence-embedding models, on both the IEDB (categorical model had a higher AUC ROC in 8 out of 11 antigens, Wilcoxon test, $P < 0.01$) and the single-cell pMHC CD8$^+$ T-cell data set (categorical model had a higher AUC ROC in 8 out of 8 antigens, Wilcoxon test, $P < 0.01$; Fig 3D and E). We conclude that the previously proposed antigen sequence-embedding models are currently suboptimal for binding prediction on seen antigens. Moreover, the analysis of seen antigens does not suggest that antigen-embedding models can learn representations of antigen sequences that allow for generalization in the antigen space.

Secondly, we tested the ability of sequence-embedding models to generalize to held-out antigens that are not contained in the training data. This task cannot be performed with models that treat antigens as categories. Firstly, we trained models on a subset of high-frequency antigens from IEDB and tested on low-frequency antigens from IEDB (Appendix Fig S5A, average of the top mean AUC ROC by antigen of 0.79). Secondly, we used a subset of observations of VDJdb with antigens not overlapping to IEDB as a test set (Appendix Fig S5B, average of the top mean AUC ROC by antigen of 0.86). Thirdly, we trained models on IEDB and tested on not overlapping antigens from the single-cell pMHC CD8$^+$ T-cell data (Appendix Fig S5C, average of the top mean AUC ROC by antigen of 0.59). While binding could be predicted for a few held-out antigens, the variation in prediction success across antigens and data sets was very large on the IEDB and VDJdb hold out scenarios. In the single-cell hold out (Appendix Fig S5C), in which we had sufficient data to assess prediction properly for each antigen, the predictivity was not very high (average AUC ROC 0.59). Thus, we cannot find evidence in the current TCR databases that extrapolation in the antigen space is possible based on current numbers of sampled antigens, in accordance with previous findings (preprint: Jurtz *et al*, 2018).

In summary, we do not find evidence that supports the usage of antigen-embedding models as they are outperformed by categorical models in the task of predicting antigens contained in the training data and because there is not enough antigen diversity in the available training data to fit models that are able to generalize to unseen antigens.

### Imputation of antigen specificity of T cells adds phenotypic information to single-cell studies

We showed that antigen specificity can be predicted based on TCR sequences from single-cell data. The inclusion of pMHC binding detection in an experiment increases the sequencing and reagent costs compared with experiments involving CDR3 sequencing only; this will be especially pronounced in assays with many different antigens. However, antigen specificity is a layer of phenotypic information that adds to single-cell RNA-seq embeddings and can be used to relate activation states and cell types to specific disease-causing agents. The model classes shown here can be used to impute antigen specificity based on the CDR3 sequence only. Accordingly, pre-trained specificity-predicting models may serve as an alternative to including pMHCs in T-cell assays. All models discussed above can be used for the purpose of imputation. We found that the imputation of antigen specificity can give interpretable results in T-cell subpopulations identified based on the transcriptome (Fig 4): The observed labels are enriched in sub-regions of the transcriptome space (Fig 4A and C), which can be recovered in multiple cases based on the predicted labels (Fig 4B and D). This implies that cell states can be interpreted based on imputed specificity labels. In this scenario, one encounters the case of held-out donors. We showed above that prediction performance is strongly increased if donors are modeled (Fig 1C). Prediction to unseen donors requires the MHC alleles to be modeled directly; this requires larger patient cohorts than given in this study, though, and will be a focus of future research.

## Discussion

Our results quantify the benefit of jointly modeling the TCR α- and β-chains while accounting for single-cell variability through cell- and donor-specific covariates for the prediction of T-cell specificity. Most importantly, we found that models that treat antigens as categorical outcome variables outperform those that model the TCR and antigen sequences jointly. Our results suggest that T-cell specificity can be predicted in an HLA genotype-specific fashion and thereby pave the way for research and development on all HLA types, beyond the commonly investigated type HLA-A*02:01. Here, we modeled donor rather than explicitly modeling MHC alleles. In the future, one might directly use one-hot encoded MHC alleles as predictors when larger patient cohorts become available. The issue of MHC allele modeling is much simplified if pMHC panels are considered in isogenic mouse models only, which may be an important scenario for mouse-based single-cell immunology research. We showed that generalization to unseen antigens with antigen sequence-embedding models is currently challenging. However, these models will become more important as the diversity of assayed antigens increases. The models and analysis presented here can serve as a starting point for such studies in the future. Lastly, we showed that pMHC counts can be modeled as a measure of the strength of dextramer to T-cell binding and that multi-task models outperform single-task models in this setting, facilitating the integration of large pMHC panels in single experiments.

T-cell specificity complements standard immunological single-cell RNA-seq studies and can be used to uncover subpopulations that are expected to be activated during disease or used as an indicator of the presence of an antigen in a tissue. Consequently, we propose the computational imputation of T-cell specificity as

an important tool for immunologically focused single-cell RNA-seq experiments. Here, we chose a very conservative exclusion of putative doublet T cells that could be improved in the future based on the transcriptome-derived and the TCR sequence-derived doublet likelihoods of each observation. Imputation will reduce the number of pMHC species in experiments by allowing antigen prioritization or may entirely replace the pMHC reagents in this workflow. In addition to the economic value of this imputation, it will also offer unbiased specificity metrics that are not liable to errors in the pMHC panel choice. Such predictive models can also be directly applied to immunophenotyping by screening for TCRs that interact with known viral or cancer neoepitopes, enabling the characterization of a patient's immunological state and the stratification of subpopulations that are amenable to antigen-specific immunotherapies. Continuous T-cell binding strength models would permit the possibility of rational *in silico* TCR design, accelerating the development of TCR-based biologics.

# Materials and Methods

## Reagents and Tools table

| Reagent/ Resource | Reference or Source | Identifier or Catalog Number |
|---|---|---|
| **Software** | | |
| *python* v3.7 | https://www.python.org/ | |
| scanpy v1.4 | https://pypi.org/project/scanpy/ | |
| tensorflow v2.0.1 | https://pypi.org/project/tensorflow/ | |

## Methods and Protocols

### General note on data sets

In this study, we worked on data sets from public databases IEDB (Vita *et al*, 2019) and VDJdb (Shugay *et al*, 2018) and on a public data set from a single-cell pMHC-based T-cell specificity experiment (10x Genomics, 2019). IEDB and VDJdb contain pairs of binding T-cell receptors (TCRs) and antigens. In the single-cell experiment, cells were first treated with barcoded pMHCs and were then physically separated into droplets in a microfluidics setup. pMHCs captured in these droplet and T-cell receptor sequences associated with the captured cells are barcoded with a droplet-specific sequence so that both can be mapped to a single observation after sequencing (10x Genomics, 2019). Accordingly, one can obtain not only a list of bound TCRs and antigens but also pMHC counts for each TCR. These counts can be discretized into binding events and "spurious" binding or can be directly modeled as proposed in the main text. Importantly, one can easily establish the identity of multiple binding antigens to a single TCR sequence based on such pMHC counts. Two of the four donors (donors 1 and 2) were HLA-A*02:01 (10x Genomics, 2019), which was also the HLA type selected for in the IEDB and VDJdb samples. A detailed description of the HLA types and pMHC types used in this study is provided elsewhere (10x Genomics, 2019).

### Statistics

We present *P*-values for selected model performance comparisons. These *P*-values were computed on the comparison of two sets of performance metrics. We used Welch's *t*-test if we compared two sets of performance metrics from two separate cross-validation sets, which is equivalent to the case of both sets sharing all model hyper-parameters other than cross-validation partition. We used the Wilcoxon test if we compared metrics across sets of models that vary in hyper-parameters, as one would no longer expect a unimodal performance metric distribution in these cases.

### Feed-forward network architectures

Here, we describe proposed architectures of the models that predict antigen specificity of a T-cell receptor (TCR) based on the CDR3 loop of both α- and β-chains and on cell-specific covariates. Note that specificity-determining influences of CDR1 and CDR2 loops (Cole *et al*, 2009; Madura *et al*, 2013; Stadinski *et al*, 2014) and distal regions (Harris *et al*, 2016a,b) have also been demonstrated, but were not measured in the single-cell pMHC assay. All networks presented contain an initial amino acid embedding, a sequence data embedding block, and a final densely connected layer block.

### Amino acid embedding

The choice of initial amino acid embedding may impact data and parameter efficiency of the model and therefore may impact the predictive power of models trained on data sets that are currently available. We used one-hot encoded amino acid embeddings, evolutionary substitution-inspired embeddings (BLOSUM), and learned embeddings. The learned embeddings were a $1 \times 1$ convolution on top of a BLOSUM encoding and were prepended to the sequence model layer stack. Here, channels are the initial amino acid embeddings (we chose BLOSUM50) and filters are the learned amino acid embedding. This learned embedding can reduce the parameter size of the sequence model layer stack. All fits presented in the manuscript other than in Appendix Fig S1 are based on such a learned embedding with five filters. We anticipate that sequence-based embeddings will gain relevance in the context of extrapolation across antigens in the future. Here, parameter efficiency in the sequence models will play an important role and the $1 \times 1$ convolution presented here is an intuitive first step in this direction.

### Sequence data embedding

We screened multiple layer types in the sequence data embedding block: recurrent layers (bidirectional GRU and LSTM), self-attention, convolutional layers (simple convolutions and inception-like), and densely connected layers as a reference. Recurrent layer types and self-attention layers were previously useful for modeling language (Vaswani *et al*, 2017) and epitope (Wu *et al*, 2019) data. Convolutional layer types have been useful for modeling epitope (Han & Kim, 2017; Vang & Xie, 2017) and image (Szegedy *et al*, 2015) data. The sequence model layers retain positional information in subsequent layers and can thereby build an increasingly abstract representation of the sequence. To achieve this on recurrent networks, we chose the output of a layer to be a position-wise network state which results in an output tensor of size (batch, positions × 2, output dimension) for a bidirectional network. This position-wise encoding occurs naturally in self-attention and convolutional networks. We did not use feature transforms with positional signals

(Vaswani *et al*, 2017) on the self-attention networks, so that the network has no knowledge of the original sequence-structure but can still retain inferred structure in subsequent layers. We presented models fit on the CDR3 loop of both α- and β-chains of the TCR (Fig 1B) and models fit on the CDR3 loop of the β-chain and the antigen sequence (Fig 3B). In both cases, we needed to integrate two sequences. To this end, we either used separate sequence-embedding layer stacks for each sequence (all models presented in Fig 1 and models indicated as "separate" in Fig 3) or by appending the two padded sequences and using a single sequence-embedding layer stack (models indicated as "concatenated" in Fig 3). We reduced the positional encoding to a latent space of fixed dimensionality in the last sequence-embedding layer of recurrent networks by the emitted state of the model on the last element of the sequence in each direction. This last layer allows usage of the same final dense layers independent of input sequence length. Convolutional and self-attention networks were not built to be independent of sequence length. We did, however, pad the input sequences to mitigate this problem on the data handled in this paper. We used a residual connection across all sequence-embedding layers. Further layer-specific hyper-parameters can be extracted from the code supplied with this manuscript (Dataset EV1 and EV2).

### Final densely connected layers

We fed the activation generated in the sequence-embedding block into a dense network that can integrate the sequence information with continuous or categorical donor- and cell-specific covariates. We modeled the binding event as a probability distribution over two states (bound and unbound) and compute the deviation of the model prediction from observed binding events via cross-entropy loss. Firstly, one can use such models to predict binding events on a single antigen represented as a single output node with a sigmoid activation function. Secondly, one can model a unique binding event among a panel of antigens with a vector of output nodes (one for each antigen and one node for non-binding) which are transformed with a softmax activation function.

### Covariate processing

We set up a design matrix inspired by linear modeling to use as a covariate matrix. We modeled the donor as a categorical covariate, resulting in a one-hot encoding of the donor. We modeled total counts, negative-control pMHC counts, and surface protein counts as continuous covariates. We $\log(x + 1)$-transformed negative-control pMHC counts and surface protein counts to increase the stability of training. We modeled total counts as the total count of mRNAs per cell divided by the mean total count.

### Training, validation, and test splits

We used training data to compute parameter updates, validation data to control overfitting, and test data to compare models across hyper-parameters. Model training was terminated once a maximum number of epochs were reached or if the validation loss was no longer decreasing. In the latter case, the model with the lowest validation in a sliding window of $n$ epochs until the last epoch was chosen; $n$ is given in the grid search scripts (Dataset EV3). The model metrics presented in this manuscript are metrics evaluated on the test data for models selected on cross-entropy

(categorical binding prediction) or mean-squared log error (dextramer count prediction) of the validation data. We provide training curves for all models that contributed to panels in this manuscript in Dataset EV3.

### Optimization

We used the ADAM optimizer throughout the manuscript for all models. We used learning rate schedules that reduce the learning rate at the time of training once plateaus in the validation metric are reached. The initial learning rate and all remaining hyper-parameters (batch size, number of epochs, patience, steps per epoch) were varied as indicated in the grid search hyper-parameter list.

### Model fitting objectives

We chose cross-entropy loss on sigmoid- or softmax-transformed output activation values to train models that predict binarized binding events and mean-squared logarithmic error (msle) on exponentiated output activation values for models that predict continuous (count) binding affinities.

### Performance metrics

We used AUC ROC, F1 scores, false-negative rates, and false-positive rates in the study to evaluate models that predict binding probabilities. AUC ROC is useful if the observations cover the full range of classification thresholds and is useful because it provides a measure that summarizes all scalar classification thresholds. F1 scores can always be used to evaluate a classifier but rely on a strict threshold. We used AUC ROC where possible but complemented with F1 scores if the AUC ROC score may suffer from a disjointed support of test data set on the classification threshold. False-negative and false-positive rates are used in Appendix Fig S4 to emphasize how models trained on single-cell data generalize to data from IEBD and VDJdb in both the negative and the positive classes separately. We used the $R^2$ to evaluate the performance of models that predicted pMHC counts (positive integer space).

### Single-cell immune repertoire (CD8[+] T cell) data processing
#### Primary data processing

We downloaded the full data of all four donors from another study (10x Genomics, 2019). All data processing for each model fit is documented in the package code (Dataset EV1) and grid search scripts (Dataset EV2). The number of T-cell clonotypes per antigen varied drastically between the order of $10^0$ and $10^4$ (Appendix Fig S3A and B). Subsequently, we selected the eight most common antigens (ELAGIGILTV, GILGFVFTL, GLCTLVAML, KLGGALQAK, RLRAEAQVK, IVTDFSVIK, AVFDRKSDAK, RAKFKQLL) for categorical panel model fits to avoid issues with class imbalances. We used the binarized binding event prediction by the authors of the data set (10x Genomics, 2019; labeled "*_binder" in the files "*_binarized_matrix.csv") as a label for prediction. For the continuous case, in which we predicted pMHC counts, we chose the corresponding count data columns in the same file. Next, we performed multiple layers of observation filtering: (i) doublet removal, (ii) clonotype down-sampling, and (iii) class down-sampling. It was previously shown that doublets, namely, droplets containing two cells targeted with the same barcode, which cannot be distinguished in

downstream analysis steps, tend to be enriched in subsets of transcriptome-derived clusters (Wolock *et al*, 2019). We propose using the number of reconstructed TCR chain alleles to identify potential doublets and demonstrate that the so characterized doublets are indeed enriched in a particular cluster in each donor (Appendix Fig S2A–D). There are cells that have two active alleles for either TCR chain, but these cannot be easily separated from doublets that arise in the cell separation process. To avoid bias of the presented results by potential cellular doublets, we chose to exclude all cells showing more than one allele for either the α- or the β-chain. We further investigated the overall contribution of potentially ambient molecules that give rise to all observed T cells and found that high-frequency chains do not dominate the overall signal (Appendix Fig S2E and F). This analysis presents an upper bound to the impact of ambient molecules on this experiment as evolutionary effects probably also contribute to over-representation of particular chain sequences. Subsequently, we removed all cellular barcodes that contain more than one α- or β-chain as mature CD8$^+$ T cells are expected to only have a single functional α- and β-chain allele. Next, we down-sampled each clonotype to a maximum of 10 observations to avoid biasing the training or test data to large clones. Here, we used clonotypes as defined by the authors of the data set in the files "*_clonotypes.csv" (10x Genomics, 2019). Lastly, we down-sampled the larger class to a maximum of twice the size of the smaller class when predicting a binary binding event for a single antigen. We did not perform this last step on multiclass and count prediction scenarios. We padded each CDR3 sequence to a length of 40 amino acids and concatenated these padded chain observations to a sequence of length 80 for models that were trained on both chains. We performed leave-one-donor-out cross-validation on models that did not take the donor identity as a covariate. We sampled 25% of the full data clonotypes and assigned all of the corresponding cells to the test set for all models that did use the donor covariate. The latter case yielded 68,716 clonotypes and 91,495 cells across all four donors. All cross-validations shown across different models are based on threefold cross-validation with seeded test–train splits resulting in the same split across all hyper-parameters. We present an analysis of the clonotype diversity encountered in this data set in Appendix Fig S6.

### Binarization of single-cell pMHC counts into bound and unbound states

We used the binarization described in the original publication (10x Genomics, 2019) for the raw counts to receive binary outcome labels: A total pMHC UMI count larger than 10 and at least five times as high as the highest observed UMI count across all negative-control pMHCs was required for a binding event. If more than one pMHC passed these criteria, the pMHC with the largest UMI count was chosen as the single binder.

### Test set assembly for models fit on IEDB data

This section describes how the test described in Fig 3E and Appendix Fig S5C was prepared. The cells were filtered as described above. We then extracted one binding TCR-antigen pair per cell from this list. We used the remaining TCR-antigen pairs as validated negative examples and down-sampled these to the number of positive observations to maintain class balance. All cross-validations shown across different models are based on threefold cross-validation with seeded test–train splits resulting in the same split across all hyper-parameters.

### IEDB data processing
#### Primary processing
We downloaded the data from the IEDB website (Vita *et al*, 2019) with the following filters: linear epitope, MHC restriction to HLA-A*02:01 and organism as human and only human. This yielded a list of matched TCR (mostly β-chain CDR3s) with bound antigens. We assigned TCR sequences to a single clonotype if they were perfectly matched and down-sampled all clonotypes to a single observation. We only extracted the β-chain and CDR3 sequences to a length of 40 amino acids. We padded the antigen sequences to a length of 25 amino acids. We sampled 10% of all observations as a test set. We generated negative samples for both training and test sets separately by generating unobserved pairs of TCR and antigens. Here, we assumed that all TCRs bind a unique antigen out of the set of all antigens present in the database so that any other pairing would not result in a binding event. This procedure yielded 9,697 observations for both the positive and the negative sets before the train–test split from 71 antigens.

### Test set assembly for models fit on IEDB data

This section describes how the test depicted in Appendix Fig S5A was prepared. To explore the ability of antigen-embedding Tcell-Match models to generalize to unseen antigens, we fit such a model on the subset of high-frequency antigens of IEDB with at least five unique TCR sequences and tested the models on the remaining antigens. All cross-validations shown across different models are based on threefold cross-validation with seeded test–train splits resulting in the same split across all hyper-parameters.

### VDJdb data processing
#### Primary processing
We provided an exploratory analysis of this data set in Appendix Fig S3 "exploration_vdjdb_data.*". We downloaded the data from the VDJdb (Shugay *et al*, 2018) website with the following filters: Species: human, Gene (chain): TRB, MHC First chain allele(s): HLA-A*02:01. This yielded 3,964 records from 40 antigens. We assigned TCR sequences to a single clonotype if they were perfectly matched and down-sampled all clonotypes to a single observation. We only extracted the β-chain and CDR3 sequences to a length of 40 amino acids. We padded the antigen sequences to a length of 25 amino acids.

### Test set assembly from VDJdb for models fit on IEDB data

This section describes how the test depicted in Fig 3D and Appendix Fig S5B was prepared. We sub-selected observations with matching or non-matching antigens with respect to the training set depending on the application (described in the figure caption or main text). All cross-validations shown across different models are based on threefold cross-validation with seeded test–train splits resulting in the same split across all hyper-parameters.

## Data availability

The data sets and computer code produced in this study are available in the following databases:

- Modeling python package (*TcellMatch*) and analysis scripts: GitHub (https://github.com/theislab/tcellmatch). Model fits are available in Dataset EV4.

**Expanded View** for this article is available online.

## Acknowledgments

We would like to thank Dr. Mike Stubbington for fruitful discussions on the topic of predicting T-cell specificity. D.S.F. acknowledges support from a German Research Foundation (DFG) fellowship through the Graduate School of Quantitative Biosciences Munich (QBM) [GSC 1006 to D.S.F.] and by the Joachim Herz Stiftung. B.S. acknowledges financial support from the Postdoctoral Fellowship Program of the Helmholtz Zentrum München. F.J.T. acknowledges financial support from the Graduate School QBM, the German Research Foundation (DFG) within the Collaborative Research Centre 1243, Subproject A17, by the Helmholtz Association (Incubator grant sparse2big, grant #ZT-I-0007), by the BMBF grant #01IS18036A, and grant #01IS18053A and by the Chan Zuckerberg Initiative DAF (advised fund of Silicon Valley Community Foundation, 182835). We thank the Center for Information Services and High Performance Computing (ZIH) at TU Dresden for generous allocations of computer time.

## Author contributions

DSF and YW implemented the modes and performed the analysis. DSF, BS, and FJT wrote the manuscript.

## Conflict of interest

F.J.T. reports receiving consulting fees from Roche Diagnostics GmbH and Cellarity Inc., and ownership interest in Cellarity Inc.

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
