## [Review Process File · Molecular Systems Biology]

Predicting antigen-specificity of single T cells based on TCR CDR3 regions

David Fischer, Yihan Wu, Benjamin Schubert, and Fabian Theis

DOI: [10.15252/msb.20199416](https://doi.org/10.15252/msb.20199416)

Corresponding author(s): Fabian Theis (Fabian.theis@helmholtz-muenchen.de)

Review Timeline:

Submission Date:	16th Dec 19
Editorial Decision:	2nd Mar 20
Revision Received:	2nd Jul 20
Editorial Decision:	3rd Jul 20
Revision Received:	13th Jul 20
Accepted:	22nd Jul 20

Editor: Maria Polychronidou

Transaction Report:

The reviewers' comments and authors' responses are not available with this article, as the initial review process took place with another journal.

Thank you again for submitting your work to Molecular Systems Biology. We sent the manuscript to reviewer #1 from the previous journal, and to a second reviewer, who unfortunately never sent us their comments despite a series of reminders. We have now finally heard back from reviewer #1, who as you will see below thinks that the performed revisions have addressed the previously raised concerns and is supportive of publication.

Reviewer #1 only recommends going through the text and making sure the language is accurate and the manuscript reads well. On a more editorial level, we would ask you to address the following points in your revision:

REFEREE REPORTS

Reviewer #1:

The authors have made an extensive effort to respond to my previous comments and to the comments from other reviewers. The paper has improved significantly and I do not have any further comments that are specific to any changes that need to be made to the scientific content. There are still some language edits that need to be made throughout the manuscript, in my opinion, prior to publication.

Point-by-point response to the reviewers' comments

Predicting antigen-specificity of single T-cells based on TCR CDR3 regions

David S. Fischer^{1,2}, Yihan Wu¹, Benjamin Schubert^{1,3}, Fabian J. Theis^{1,2,3,+}

¹Institute of Computational Biology, Helmholtz Zentrum München, 85764 Neuherberg, Germany

²TUM School of Life Sciences Weihenstephan, Technical University of Munich, 85354 Freising, Germany

³Department of Mathematics, Technical University of Munich, 85748 Garching bei München, Germany

+ Corresponding author: fabian.theis@helmholtz-muenchen.de

In the following, we present our response to the reviewers comments at the previous journal. We give **comments (black)**, **point-by-point answers (green)** to the questions and in parts **copy parts of the text or specific panels (beige)**, which directly correspond to comments or reference to them.

Reviewer #1:

The authors have made an extensive effort to respond to my previous comments and to the comments from other reviewers. The paper has improved significantly and I do not have any further comments that are specific to any changes that need to be made to the scientific content. There are still some language edits that need to be made throughout the manuscript, in my opinion, prior to publication.

We would like to thank the reviewer for acknowledging our efforts and worked on the writing style.

Thank you for sending us your revised manuscript. We are satisfied with the modifications made and I am glad to inform you that your manuscript is now suitable for publication.

Before we formally accept the manuscript, we would ask you to address a few remaining editorial issues.

Corresponding Author Name: Fabian J Theis

Journal Submitted to: Article

Manuscript Number: MSB-19-9416R